# Identification and Characterization of Two Novel Noda-like Viruses from Rice Plants Showing the Dwarfing Symptom

**DOI:** 10.3390/v14061159

**Published:** 2022-05-27

**Authors:** Yi Xie, Shuai Fu, Li Xie, Yaqin Wang, Mengji Cao, Xueping Zhou, Jianxiang Wu

**Affiliations:** 1State Key Laboratory of Rice Biology, Institute of Biotechnology, Zhejiang University, Hangzhou 310058, China; 11616061@zju.edu.cn (Y.X.); fushuai@zju.edu.cn (S.F.); wangyq0219@126.com (Y.W.); 2Analysis Center of Agrobiology and Environmental Sciences, Zhejiang University, Hangzhou 310058, China; hyena_xieli@163.com; 3National Citrus Engineering and Technology Research Center, Citrus Research Institute, Southwest University, Beibei, Chongqing 400712, China; 4State Key Laboratory for Biology of Plant Diseases and Insect Pests, Institute of Plant Protection, Chinese Academy of Agricultural Sciences, Beijing 100193, China

**Keywords:** rice, nodavirus, noda-like virus, reverse genetic system, RNA silencing suppressor

## Abstract

Nodaviruses are small bipartite RNA viruses and are considered animal viruses. Here, we identified two novel noda-like viruses (referred to as rice-associated noda-like virus 1 (RNLV1) and rice-associated noda-like virus 2 (RNLV2)) in field-collected rice plants showing a dwarfing phenotype through RNA-seq. RNLV1 genome consists of 3335 nt RNA1 and 1769 nt RNA2, and RNLV2 genome consists of 3279 nt RNA1 and 1525 nt RNA2. Three conserved ORFs were identified in each genome of the two novel viruses, encoding an RNA-dependent RNA polymerase, an RNA silencing suppressor, and a capsid protein, respectively. The results of sequence alignment, protein domain prediction, and evolutionary analysis indicate that these two novel viruses are clearly different from the known nodaviruses, especially the CPs. We have also determined that the B2 protein encoded by the two new noda-like viruses can suppress RNA silencing in plants. Two reverse genetic systems were constructed and used to show that RNLV1 RNA1 can replicate in plant cells and RNLV1 can replicate in insect Sf9 cells. We have also found two unusual peptidase family A21 domains in the RNLV1 CP, and RNLV1 CP can self-cleave in acidic environments. These findings provide new knowledge of novel nodaviruses.

## 1. Introduction

Nodaviruses are small single-stranded positive-sense RNA viruses belonging to the family *Nodaviridae* [1]. Virions of nodaviruses are non-enveloped and spherical with icosahedral symmetry. The diameters of nodavirus virions range from 25 nm to 33 nm. The genome of nodaviruses is bipartite (i.e., RNA1 and RNA2) and contains a total of ~4500 nucleotides (nt). Both viral RNA1 and RNA2 are methylated-capped at the 5′-terminus and lack a poly(A) tail at the 3′-terminus [1]. Nodavirus RNA1 encodes an RNA-dependent RNA polymerase (RdRp, also known as protein A) [1]. Nodavirus RNA2 encodes a capsid protein (CP, also known as protein α) [1]. During RNA replication, a subgenomic RNA3 is synthesized from the 3′-end of RNA1 using a premature termination strategy [1]. This subgenomic RNA3 is not packaged in virions and encodes one or two small proteins (i.e., protein B1 and B2) [1]. The protein B1 is reported for some known nodaviruses and its function in virus infection is largely unknown [1]. The protein B2 is encoded by all nodaviruses, and functions as an RNA silencing suppressor [1].

The name Noda is from the first reported nodavirus, Nodamura virus (NoV), isolated from mosquitos (*Culex tritaeniorhynchus*) in Nodamura, Japan [2]. Several similar viruses were later reported in insects [3,4,5,6], and other related viruses were found in fishes [7,8,9,10,11,12,13]. Based on the genetic diversity among nodavirus RNA2s, nodaviruses are classified into two genera: *Alphanodavirus* and *Betanodavirus*, infecting insect and fish, respectively [1]. More nodaviruses have recently been identified in other organisms [1]. For example, several research groups have found nodaviruses in different species of shrimp and prawn [14,15,16]. In 2013, a third group of nodaviruses, namely *Gammanodavirus*, has been proposed according to the distant phylogenetic relationships between their CPs and the CPs of known nodaviruses [17]. Nematodes are also known as the hosts of some noda-like viruses that possess unique genome features [18,19,20]. Moreover, some nodaviruses have been found in mammals or their fecal samples [21,22,23,24,25]. Although nodaviruses are currently considered to be animal viruses, some noda-like viruses have also been detected in plant samples through RNA-seq technologies [26,27].

Nodaviruses can be transmitted vertically and horizontally in their hosts [1]. Betanodaviruses have been reported to cause nervous necrosis in fish, while several nodaviruses can induce muscle necrosis in shrimp and prawn, leading to significant losses in aquaculture worldwide [1,28]. To date, alphanodaviruses have not been reported to cause significant threats to the global economy [28]. Due to their small genome size and efficient replication in many hosts, several alphanodaviruses, especially Flock House virus (FHV), have been modified and successfully used as excellent models for the studies of positive-strand RNA virus infections in their hosts [29,30,31,32].

Virus-infected rice plants often show severe plant stunting, shoot malformation, and reduction in grain production [33,34,35]. Rice viruses currently known to occur in rice fields in China include rice stripe virus (RSV), rice black-streaked dwarf virus (RBSDV), southern rice black-streaked dwarf virus (SRBSDV), rice ragged stunt virus (RRSV), rice gall dwarf virus (RGDV), rice stripe mosaic virus (RSMV), and rice dwarf virus (RDV). These viruses are transmitted in fields by planthoppers or leafhoppers and can replicate in their insect vectors [33,34,35,36]. The most common strategies used to control rice virus diseases in China are cultivation of virus-resistant varieties, applications of insecticides to prevent virus transmission by insects, and timely detection and removal of virus-infected plants from rice fields [33,34,35,37].

Several serological and molecular techniques, such as enzyme-linked immunosorbent assay (ELISA), molecular hybridization, polymerase chain reaction (PCR), and reverse transcription-PCR (RT-PCR), have been successfully used to detect known viruses in rice fields [37,38]. For unknown viruses, however, the newly developed high-throughput RNA sequencing (RNA-seq) and bioinformatics have been proved to be crucial. Using these new technologies, many new viruses, including several new nodaviruses, have been found in different organisms or environments, which enriches our understanding of the viruses and provides technical support for monitoring viral diseases [37,38,39,40]. The authors have discovered a novel picornavirus in rice plants using RNA-seq in 2021 [41].

In this study, we identified two novel noda-like viruses in field-collected rice plants through RNA-seq, and named these two viruses as rice-associated noda-like virus 1 (RNLV1) and rice-associated noda-like virus 2 (RNLV2). RNLV1 RNA1 and RNA2 contain 3335 and 1769 nt, respectively, while RNLV2 RNA1 and RNA2 contain 3279 and 1525 nt, respectively. Genomes of RNLV1 and RNLV2 contain three conserved ORFs which encode an RNA polymerase, a capsid protein (CP), and an RNA silencing suppressor. Prediction of protein domains and evolutionary analysis indicate that both novel viruses are different from the known nodaviruses, especially their CPs. Through reverse genetic analysis, we have found that RNLV1 RNA1 can replicate in plants, and RNLV1 can replicate in insect Sf9 cells. We have also found two unusual peptidase family A21 domains in the RNLV1 CP, and RNLV1 CP can self-cleave in acidic environments.

## 2. Materials and Methods

### 2.1. Sample Collection and Plant Growth

Rice plants showing dwarf phenotypes were collected from rice fields in the Zhejiang Province in 2016 and in Shanghai City in 2019. The collected leaf samples were cleaned with water, frozen in liquid nitrogen, and then stored at −80 °C until further analyses. *Nicotiana benthamiana* and rice (*Oryza sativa* ssp. japonica) plants were grown inside a growth chamber maintained at 25 °C (day) and 18 °C (night), 60% relative humidity, and a 16 h (light) and 8 h (dark) photoperiod. Seeds of the GFP transgenic 16c *N. benthamiana* line were kindly provided by Professor David C. Baulcombe (University of Cambridge, Cambridge, UK).

### 2.2. RNA Sequencing (RNA-Seq) and De Novo Assembly

Total RNA was extracted from the collected rice leaf samples using the MiniBEST Plant RNA Extraction Kit (TaKaRa, Tokyo, Japan) and then used for RNA-seq. Briefly, after removal of ribosomal RNAs, RNA libraries were constructed using the TruSeq RNA Sample Prep Kit as instructed (Illumina, San Diego, CA, USA) and then sequenced on an Illumina HiSeq X-ten platform (Biomarker Technologies, Beijing, China). Adapters were trimmed from the paired-end raw reads, and low-quality reads were removed using the CLC Genomics Workbench 9.5 software (QIAGEN, Germantown, MD, USA). High quality clean reads were mapped to the rice genome database (Available online: https://www.ncbi.nlm.nih.gov/taxonomy/39947 (accessed on 8 October 2019)) to remove host sequences. The remaining reads were assembled de novo using the Trinity program (Available online: http://trinityrnaseq.github.io/ (accessed on 9 October 2019)). The assembled contigs were used to search viral sequences in the GenBank using the Basic Local Alignment Search Tools (BLASTn and BLASTx, Available online: https://blast.ncbi.nlm.nih.gov/ (accessed on 13 October 2019)). 

### 2.3. Reverse Transcription Polymerase Chain Reaction (RT-PCR), Rapid Amplification of cDNA Ends (RACE), and Junction PCR

Total RNA was extracted from the plant or insect cell samples using the RNAiso Plus as instructed (TaKaRa, Tokyo, Japan). For initial detections of viruses in the collected rice samples, we performed reverse transcription (RT) using the ReverTra Ace qPCR RT Master Mix supplemented with a gDNA Remover (Toyobo, Osaka, Japan) followed by PCR using the Green Taq Mix (Vazyme, Nanjing, China) and PCR primers designed according to the assembled contigs. For amplification of long fragments, RT was performed using M-Mlv (Takara) and fragment-specific reverse primers followed by PCR using Phanta DNA polymerase (Vazyme) and specific primer pairs. The 5′-RACE was performed using the SMARTer RACE 5′/3′ Kit as instructed (Takara). The 3′-RACE was performed through addition of a poly(A) tail to the RNA fragments using the Poly(A) polymerase (Takara) followed by RT using M-Mlv and an oligo-dT-adapter primer. A series of nested-PCR amplification were then performed using the gene-specific primers (GSPs) and the adapter primer. The junction PCR was performed through RT using M-Mlv and the primers specific for the head-to-tail (−)gRNA fragments followed by PCR amplification of the 3′–5′ junctions using specific primers. The resulting PCR products were gel purified and cloned into the TopoTA pCE2 cloning vector (Vazyme) followed by Sanger sequencing (TsingKe Biotech Co., Beijing, China). The resulting sequences were checked manually and then assembled to produce the final viral genome sequences. Based on the assembled sequences, the full-length viral RNAs were amplified from the collected rice samples through RT-PCR, cloned, and then sequenced. For plasmid construction, we performed PCR amplification using the Phanta DNA polymerase and specific primers for viral sequences or ORFs. To analyze the virus replication, we carried out RT using the M-Mlv and reverse primers specific for the negative strand viral RNAs followed by PCR amplification using the Green Taq Mix and specific primers. All primers used in this study are listed in Appendix A.

### 2.4. Prediction of Open Reading Frames (ORFs) and Proteins

Open reading frames (ORFs) in the full-length viral sequences were predicted using the ORFfinder service (Available online: http://www.ncbi.nlm.nih.gov/orffinder (accessed on 13 November 2019)) and the published Flock House virus (FHV) sequence (NC_004146 and NC_004144) as a model. Sequences of the proteins predicted to be encoded by the ORFs were subjected to BLASTp searches against the GenBank database to identify the published viral protein sequences that shared high amino acid (aa) sequence identities (expect threshold at 0.05). Sequences of the new virus genomic RNAs and their proteins were aligned with that of the known nodaviruses using the DNAMAN7.0 software (Available online: https://www.lynnon.com/dnaman.html (accessed on 16 June 2020)). Domains in the predicted proteins were then determined using the Conserved Domain Search (CD Search, Available online: https://www.ncbi.nlm.nih.gov/Structure/cdd/wrpsb.cgi (accessed on 16 June 2020)) and the InterProScan service (Available online: http://www.ebi.ac.uk/interpro/search/sequence/ (accessed on 16 June 2020)). Protein secondary structure analysis was performed using the NetSurfP 2.0 (Available online: https://services.healthtech.dtu.dk/service.php?NetSurfP-2.0 (accessed on 16 October 2021)). Protein modeling was performed using the Phyre2 service (Available online: http://www.sbg.bio.ic.ac.uk/phyre2/ (accessed on 26 October 2021)). All sequences used in this study are listed, together with their accession numbers, in Appendix A.

### 2.5. Phylogenetic Analysis

Sequences of RdRps and CPs of the two new viruses and that of the known related viruses were individually aligned using the MUltiple Sequence Comparison by Log-Expectation (MUSLE) in MEGA X software (Available online: https://www.megasoftware.net/ (accessed on 18 February 2021)). Phylogenetic analyses of these aligned sequences were performed using the maximum likelihood (ML) algorithm in the MEGA X software (Available online: https://www.megasoftware.net/ (accessed on 18 February 2021)). The evolutionary model applied was the Jones–Taylor–Thornton (JTT) model with 1000 bootstrap replicates. All sequences used in this study are listed, together with their accession numbers, in Appendix A.

### 2.6. Plasmid Construction

The ORF1bs in the two new viruses were individually amplified, and cloned into the pGD vector pre-digested with the *BamH*I and *Sal*I restriction enzymes or into the pGR106 vector pre-digested with the *Cla*I and *Sal*I restriction enzymes using the one-step cloning kit (Vazyme). The full-length viral RNA1 and RNA2 sequence of RNLV1 was amplified and cloned into the pCB301-RZ vector pre-digested with the *Stu*I and *Sma*I restriction enzymes using the one-step cloning kit to produce pRNLV1 RNA1wt and pRNLV1 RNA2. An optimized RNLV1 RNA1 sequence (RNA1opt) was produced through introduction of synonymous mutations into the ORF1a sequence to alter all the potential RNA processing sites predicted using the Netgene2 service (Available online: https://services.healthtech.dtu.dk/service.php?NetGene2-2.42 (accessed on 29 December 2020)), de novo synthesized (GenScript, Nanjing, China), and then cloned into the pCB301-RZ vector using the one-step cloning kit to produce pCB301-RNLV1-RNA1opt. A full-length eGFP gene was then inserted into pRNLV1 RNA1wt and pRNLV1 RNA1opt to produce pRNLV1 RNA1wt-B2:GFP and pRNLV1 RNA1opt-B2:GFP, respectively, through one-step cloning (Vazyme). The RNLV1 ORF2a was amplified, and cloned into pET-28a vector pre-digested with the *BamH*I and *Sal*I restriction enzymes, pGD vector pre-digested with the *BamH*I and *Sal*I restriction enzymes, or pFASTBAC vector pre-digested with the *BamH*I and *Sal*I restriction enzymes using the one-step cloning kit (Vazyme).

### 2.7. Agro-Infiltration Assay

The pGD-based or the pCB301-based expression vectors were individually transformed into *Agrobacterium tumefaciens* strain EHA105, while the pGR106-based vectors were individually transformed into *A. tumefaciens* strain Gv3101(JS Rep) through electroporation. The transformed *A. tumefaciens* cultures were grown individually, pelleted, and then diluted in an infiltration buffer (10 mM MgCl_2_, 10 mM MES, pH 5.6, and 100 μM acetosyringone) till OD_600_ = 0.6, prior to infiltration. For co-infiltration assays, two diluted *Agrobacterium* cultures carrying two different constructs were mixed at a 1:1 (*v*/*v*) ratio and then infiltrated into the leaves of *N. benthamiana* plants with 5–7 leaves using needleless syringes.

### 2.8. In Vitro Transcription and Viral RNA Inoculation

A T7 promoter was added to the 5′-end of RNLV1 RNA1 or RNA2 cDNA through PCR. In vitro RNA transcription was performed using a commercial T7 transcription kit (Vazyme) supplemented with the cap analog (New England Biolabs, Ipswich, MA, USA). The synthesized RNAs were digested with DNaseI (Takara), then extracted using the phenol/chloroform extraction method. For plant inoculation, the same amount of RNA1 and RNA2 transcripts were mixed, diluted in an inoculation buffer containing 50 mM Tris-HCl, 250 mM NaCl, and 5 mM EDTA, pH 7.5, to reach 500 ng mixed RNA transcripts per milliliter, and then rub-inoculated to the leaves of 2–3-week-old rice plants. After rinsing with water, the inoculated plants were allowed to grow inside a growth chamber as described above. The Sf9 cell line was obtained from Gibco (Carlsbad, CA, USA). Transfection of Sf9 cells was performed using the LipoInsect transfection reagent as instructed by the manufacturer (Beyotime, Shanghai, China). Briefly, RNA1 and RNA2 transcripts (5 μg each) were mixed with the LipoInsect transfection reagent and then transfected into Sf9 cells. After 4 h incubation, the transfected reagent was replaced with a fresh culture medium followed by 72 h incubation at 28 °C. The transfected cells were collected at 3 days post inoculation (dpi) through 5 min centrifugation at 1000× *g*. The harvested cell samples were analyzed for virus replication through RT-PCR and Western blot assay. For secondary transfection, the P1-supernatant was extracted from the RNA-transfected Sf9 cells and then used to transfect freshly prepared Sf9 cells followed by 72 h incubation at 28 °C. The expression of RNLV1 CP using Bac-to-Bac Baculovirus Expression System (Gibco) was performed as instructed.

### 2.9. Protein Extraction and Western Blot Assay

The agro-infiltrated leaf patches were harvested at 3–5 dpi. To identify the RNA silencing suppressor and analyze the virus replication, total protein was extracted from the harvested leaf tissues in an extract buffer (50 mM Tris-HCl, pH 6.8, 9 M carbamide, 4.5% SDS, and 7.5% 2-mercaptoethanol). To determine whether RNLV1 CP can self-cleave, total protein was extracted under non-denatured conditions. Briefly, the infiltrated *N. benthamiana* leaves were frozen in liquid nitrogen and ground into fine powder, then homogenized in 0.05 M phosphate buffered saline (PBS, pH 7.4). The Sf9 cells transfected with pFASTBAC-RNLV1 CP were lysed in PBS with 0.5% NP-40. The homogenate of *N. benthamiana* leaves and the lysate of Sf9 cells were clarified by centrifugation. The supernatant was divided into two equal parts, and one part of the supernatant was filtered through a centrifuge filter and then diluted in an acid buffer (50 mM NaAc, 250 mM NaCl, pH 5.0). Both parts were incubated at room temperature for 1 d. Western blot assays were performed using an anti-GFP rabbit monoclonal antibody (1:8000 dilution; Abcam, Cambridge, UK) or an anti-RNLV1 CP rabbit polyclonal antibody (1:10,000 dilution) as the primary antibody, and using an anti-rabbit IgG conjugated with horseradish peroxidase (1:8000 dilution; SIGMA, Burlington, MA, USA) as the second antibody.

## 3. Results

### 3.1. Identification of Two Noda-like Viruses in Dwarf Rice Plants through RNA-Seq

In 2019, during a survey of rice viral diseases in Shanghai City, China, we found rice plants showing similar dwarfing symptoms in a paddy field of about one hectare (Figure 1a, left). In 2016, during rice field survey in Zhejiang Province, China, we found some dwarf rice plants in multiple paddy fields of approximately twenty hectares (Figure 1a, right). Rice plants showing strong stunting phenotypes were collected from these paddy fields. Analyses of these collected rice samples through RT-PCR using primers specific for the known rice viruses (RBSDV, SRBSDV, RSV, RDV, RRSV, and RSMV) in these two regions did not yield positive results (data not shown). To further investigate the causal agent(s) of these plants, we isolated total RNA from individual rice samples, combined, and then analyzed through RNA-seq. The resulting high-quality reads were assembled de novo to produce contigs (Appendix A). The contigs were then used to BLASTx search the NCBI databases. This search identified two highly abundant contigs (i.e., contig3 and contig15) obtained from the assayed rice samples collected from Shanghai. The sequence of the predicted protein from contig15 shared 43% amino acid (aa) sequence identity (79% sequence coverage) with the protein A encoded by Mosinovirus. The sequence of the predicted protein from contig3 shared 29–39% aa sequence identity (80–96% sequence coverage) with the capsid proteins (CPs) encoded by several noda-like viruses. Analysis of rice samples collected from Shanghai through RT-PCR using primers designed according to the two contigs showed that these samples did contain RNAs representing the two contig sequences. Analysis of rice plants showing normal growth using the same RT-PCR assay did not yield similar products (Figure 1b). Analysis of rice samples collected from the Zhejiang Province through RNA-seq identified two highly abundant contigs (i.e., contig3545 and contig3206). Contig3545 contained one ORF and the sequence of the predicted protein shared 50% sequence identity (94% sequence coverage) with the RdRp of Nodamura virus (NoV), while the sequence of the predicted protein from contig3206 shared about 30% sequence identity (30–56% sequence coverage) with the CPs of several unclassified nodaviruses. RT-PCR analysis of the rice samples from the Zhejiang Province using primers designed according to these two contigs showed that these assayed rice plants did contain RNAs representing these two contig sequences (Figure 1b). Based on the above results, we concluded that the assayed rice plants contained two different new noda-like viruses and named the virus from Shanghai as rice-associated noda-like virus 1 (RNLV1) and the virus from the Zhejiang Province as the rice-associated noda-like virus 2 (RNLV2).

### 3.2. Determination of Genomic Sequences of the Two New Noda-like Viruses

In this study, several initial attempts were made to join the two genomic RNAs of RNLV1 or RNLV2 from infected rice samples through RT-PCR using primers designed according to the contig sequences, but all failed. This result indicated that these two viral RNAs were two separate molecules. We then analyzed the terminal sequences of the RNAs through 5′ and 3′ RACE. The results showed that the RNA1 and RNA2 of NLRV1 shared the same 5′-terminus (5′-GAUAUAUUAUC...-3′) (Table 1). Because the 3′-ends of alphanodavirus genomic RNAs are blocked for chemical and enzymatic modifications, conventional RACE methods could not be used to amplify alphanodavirus RNAs [42,43]. Given that the negative-strand RNAs of alphanodaviruses (i.e., FHV, NoV, or Pariacoto virus (PaV)) are known to have joined 3′ and 5′-ends [6,29,44], we tried to amplify RNLV1 RNAs through RT-PCR using their negative-strand RNAs as the templates. Through terminal modification RACE, RNLV1 RNAs were found to have an additional G at their 3′-termini compared to that obtained through junction-PCR [i.e., 5′-...CCGGC(G)-3′ (RNA1) and 5′-...CUGGC(G)-3′ (RNA2)] (Table 1). In this study, we also RT-PCR amplified the junction region using the negative-strand subgenomic RNA3 as the template. The sequencing results showed that the subgenomic RNA3 was synthesized from RNA1, starting from the nucleotide position 2814. Using these results, we were able to amplify the full-length cDNAs representing RNLV1 RNA1 and RNA2, respectively. Complete genome sequences of RNLV1 have now been deposited in the GenBank under the accession numbers ON260786 (RNA1) and ON260787 (RNA2).

In this study, we also amplified the full-length RNLV2 genomic RNA1 and RNA2 and analyzed their 5′- and 3′-end sequences through RACE. The result showed that the six analyzed RNA1 clones all had GGUUUU at their 5′-ends and the six analyzed RNA2 clones had GGUAUU at their 5′-ends (Table 2). The 3′-RACE result showed that five of the nine analyzed RNA1 clones had AACGGU(A) at their 3′-ends, while the other four clones ended with AACGU(A) (Table 2). All the five RNLV2 RNA2 clones had UUUCU(A) at their 3′-ends (Table 2). Because junction PCR did not yield any expected product, we speculated that the RNA in the 2016 collected rice samples was degraded during the storage. Complete genome sequences of RNLV2 have been deposited in the GenBank under the accession numbers ON260788 (RNA1) and ON260789 (RNA2).

### 3.3. Genome Organizations of RNLV1 and RNLV2

Analysis of full-length RNLV1 RNA1 and RNA2 sequences found that these two RNAs contained 3335 and 1769 nucleotides (nt), respectively. In addition, RNA1 was found to contain two ORFs (ORF1a with 3180 nt and ORF1b with 441 nt) (Figure 2a). The ORF1a was predicted to encode a protein that shared 15.53–36.27% sequence identities with the RdRps of known viruses in the family *Nodaviridae* (Table 3). Further analysis of this predicted protein with the CD Search and InterProsScan services identified a methyltransferase (MT) domain and an RNA dependent RNA polymerase (RdRp) domain (Figure 2c) [45]. Consequently, we consider that the ORF1a encodes a viral RdRp. The RNLV1 ORF1b, a small ORF that overlaps partially with the ORF1a, is in the +1 frame with the RdRp-encoding ORF, as described previously for the ORF encoding B2 protein of other nodaviruses [1]. The ORF1b encoded protein shared 8.72–17.01% sequence identities with the B2 proteins of other known nodaviruses (Table 3). Because we were unable to identify any conserved domain in the RNLV1 B2 protein using the CD Search and InterProScan services, we analyzed the secondary structure of this protein using the NetSurfP2.0 service. The result showed that the RNLV1 B2 protein could form multiple helixes similar to that found in other nodavirus B2 proteins (Appendix A). Based on the results above, we named the protein encoded by ORF1b as B2 protein. RNLV1 RNA2 was found to have one ORF (ORF2a) with 1569 nt in length (Figure 2a) and encode a protein that shared 8.43–16.42% sequence identities with the capsid proteins (CPs) of known nodaviruses (Table 3). Through InterProScan service, a viral CP domain was found in the RNLV1 ORF2a encoded protein (Figure 2d), thus, we named it as RNLV1 CP. It is noteworthy that two peptidase family A21 (cl04161) domains were found in the RNLV1 CP using the CD Search (Figure 2d, Appendix A). This domain is only found in CPs of tetraviruses and as a type of endopeptidase [46,47]. The CPs of alaphnodavirus are also known as endopeptidases (EC 3.4.23.44) that can catalyze the hydrolysis of an asparaginyl bond at the C-terminus of CP during virion maturation. The CPs of alaphnodavirus have the peptidase family A6 domain (cl03372), according to the prediction using CD Search (Appendix A). In this study, we analyzed the CPs similar to RNLV1 CP retrieved from the NCBI database and found the peptidase A21 domain in most of them (Appendix A). In addition, we found that two unclassified nodaviruses (Mosinovirus (MoNV) and Lutzomyia nodavirus (LNV)) can also encode CPs with the peptidase family A21 domain (Appendix A), indicating that RNLV1 CP is a unique nodavirus CP.

RNLV2 genome is also bipartite. RNLV2 RNA1 and RNA2 are 3279 and 1525 nt in length, respectively (Figure 2b). RNA1 contains two ORFs (1a and 1b) (Figure 2b). The ORF1a was predicted to contain 3084 nt and encode a protein that shared 14.91–47.76% sequence identities with the RdRps of the known nodaviruses (Table 4). The highest pairwise identity (47.76%) was found between the RNLV2 1a protein and the RdRp of NoV. The conserved protein domains found in the nodavirus RdRps were also found in the RNLV2 1a protein through CD Search and InterProScan services (Figure 2c) [45]. Thus, we consider that the RNLV2 ORF1a encodes a viral RdRp. The predicted RNLV2 ORF1b-encoded protein shared 10.16–22.96% sequence identities with the B2 proteins of the known nodaviruses (Table 4). Prediction results using CD Search showed that this protein contained a B2 superfamily domain (cl12995) (Appendix A), protein modeling using Phyre2 also showed a B2-like model similar to that of NoV B2 (Appendix A), indicating that it is a conserved B2 protein. RNLV2 RNA2 was predicted to contain one ORF (ORF2a) with 1179 nt in length (Figure 2b) and encode a protein that shared 8.16–21.68% sequence identities with the CPs of nodaviruses (Table 4). The prediction results of CD Search and InterProScan services indicated that the protein encoded by the ORF2a contained a viral CP S domain, commonly found in the CPs of betanodaviruses and the CPs of unclassified nodaviruses (Figure 2d, Appendix A). This predicted protein, however, did not have the peptidase domain, indicating that this protein is different from the CP of RNLV1 (Table 4).

### 3.4. Phylogenetic Analyses

To determine the phylogenetic relationships between these two new noda-like viruses and the other reported nodaviruses, we conducted phylogenetic analyses using the deduced RdRp and CP aa sequences. The result showed that the RNLV1 RdRp could be clustered in a clade together with the RdRps of PaV, MoNV, and Wuhan nodavirus (WhNV) (Figure 3a). The RNLV2 RdRp could be clustered in a clade together with the RdRps of NoV, LNV, Macrobrachium rosenbergii nodavirus, and Penaeus vannamei nodavirus (Figure 3a). The RNLV1 CP was found to have a distant phylogenetic relationship with the CPs of other reported nodaviruses, similar to that reported for the CPs of MoNV [48] (Figure 3b). We speculate that RNLV1 might acquire its CP from a species beyond the family *Nodaviridae*. The RNLV2 CP was found to be clustered in a large clade together with the CPs of betanodaviruses and some unclassified nodaviruses infecting different hosts (Figure 3b), even though the RNLV2 CP shared low sequence identities with these CPs (Table 4).

The taxonomy of nodaviruses is decided according to the genetic diversities of their RNA2 segments [1]. Therefore, novel virus species will be determined if the viral RNA2 shares less than 80% sequence identity with that of the most closely related virus species, and its encoded CP shares less than 87% aa sequence identity with that of the most closely related virus species [1]. The results presented in Table 3 and Table 4 indicated that the two viruses identified in this study should be considered as new virus species. However, we are unable to place these two new viruses in any existing genus in the family *Nodaviridae*, due to the differences in viral protein sequences, host ranges, and phylogenetic relationships.

### 3.5. RNLV1 and RNLV2 B2 Proteins Function as RNA Silencing Suppressors

Although the B2 proteins of different nodaviruses share low sequence identities, they all possess RNA silencing suppression activities [1,48,49,50,51]. In this study, a conserved small ORF was predicted at the 3′-end of RNA1 in these two new viruses, similar to other nodaviruses. Furthermore, our BLASTp search result showed that RNLV2 B2 protein was similar to NoV B2. To investigate whether the RNLV1 and RNLV2 B2 proteins can also suppress RNA silencing in plants, we co-expressed mGFP5 and RNLV1 B2 or mGFP5 and RNLV2 B2 in the leaves of transgenic 16c *N. benthamiana* plants. The result showed that in the presence of RNLV1 B2 or RNLV2 B2, strong green fluorescence was observed in the infiltrated leaf regions, similar to that observed in the leaf regions co-expressing mGFP5 and the P19 of tomato bushy stunt virus (TBSV) (Figure 4a). As expected, by 5 days post agro-infiltration (dpi), the leaf regions co-expressing mGFP5 and the empty vector (EV) showed very weak green fluorescence, indicating that the expression of the GFP in the infiltrated transgenic 16c *N. benthamiana* leaves was silenced (Figure 4a). Western blot assay result confirmed that much more GFP had accumulated in the leaf tissues co-expressing mGFP5 and RNLV1 B2, RNLV2 B2, or TBSV P19, compared to that in the leaf tissues co-expressing mGFP5 and the empty vector (the negative control) (Figure 4b). In a separate study, *N. benthamiana* plants were inoculated with a PVX-based RNLV1 B2 expression vector or PVX vector. Systemic leaves of the inoculated plants showed severe leaf curling and plant stunting phenotypes followed by necrosis at 9 dpi (Figure 4c). In contrast, the plants inoculated with the PVX vector showed only mild mosaic symptoms (Figure 4c). Plants inoculated with a PVX-based RNLV2 B2 expressing vector showed a delayed disease symptom and developed necrosis in their systemic leaves by 11 dpi (Figure 4d). Thus, we propose that both RNLV1 and RNLV2 B2 proteins are RNA silencing suppressors and related to symptom determinants.

### 3.6. RNLV1 RNA1 Can Replicate in Plant Cells

It was reported that nodavirus RNA1 can replicate independently in mammalian cells [52]. To determine whether RNLV1 RNA1 can replicate in plant cells, we inserted a full-length *eGFP* gene before the stop codon (TAG) of the ORF1b (*B2* gene) to produce a plasmid pCB301-RNLV1 RNA1-B2:eGFP (referred to as pRNLV1 RNA1wt-B2:eGFP thereafter) (Figure 5a). If the viral RNA1 expresses the RdRp protein and subsequently replicate to generate subgenomic RNA3, B2:eGFP fusion protein would be expressed in agro-infiltrated *N. benthamiana* leaf cells. After infiltration of *A. tumefaciens* harboring pRNLV1 RNA1wt-B2:eGFP into *N. benthamiana* leaves, no green fluorescence was observed in the infiltrated leaves by 3 dpi. (Figure 5b, left image), indicating that this RNA1 did not replicate in *N. benthamiana* cells. We proposed that this is possibly due to that the wild-type RNA1 is targeted for mRNA processing, leading to no full-length RdRp for viral RNA replication. To overcome this issue, we optimized the sequence of ORF1a through introducing synonymous mutations to mutate the predicted mRNA processing sites (Appendix A) and produced a plasmid pCB301-RNLV1 RNA1opt-B2:eGFP (referred to as pRNLV1 RNA1opt-B2:eGFP thereafter). Infiltration of *A. tumefaciens* harboring this new plasmid to *N. benthamiana* leaves yielded strong green fluorescence in the infiltrated leaves by 3 dpi (Figure 5b, right image). The result of RT-PCR using the specific primer pairs for the negative strand of RNLV1 RNA1 found that the specific gene segment was amplified from the leaves agro-infiltrated with pRNLV1 RNA1opt-B2:eGFP (Figure 5c), indicating RNLV1 RNA1opt did replicate in the infiltrated leaves. In addition, the accumulation of B2:eGFP fusion protein in the infiltrated leaves was confirmed through Western blot assay (Figure 5d).

### 3.7. Development of a RNLV1 Reverse Genetic System

Multiple reverse genetic systems have been developed and used to study nodavirus infections in suitable host cells [6,44,52,53,54,55]. In this study, we also tried to develop a reverse genetic system of RNLV1 in insect Sf9 or rice plant cells. Initially, RNLV1 RNA1 and RNA2 were in vitro transcribed from two expression vectors using a T7 transcription kit supplemented with the cap analog. The resulting RNLV1 RNA1 and RNA2 transcripts were 1:1 (*v*/*v*) mixed and then inoculated to insect Sf9 cells or rice plants. Sterile water was used as the negative control. Next, P1-supernatant was extracted from the viral RNA-transfected Sf9 cells and then used to inoculate fresh Sf9 cells or rice plants. Abnormal cellular changes including deformity and rupture were observed in the P1-supernatant-inoculated Sf9 cells by 2 dpi (data not shown). Lysate of supernatant-inoculated Sf9 cells was also prepared and examined under an electron microscope. The result showed that the lysate sample contained spherical virions of about 33 nm in diameter (Figure 6a), similar to that reported for nodaviruses [1]. Infection of RNLV1 in Sf9 cells or rice plants was then analyzed through RT using primers specific for the negative strand RNLV1 RNA1 or RNA2 followed by PCR. The results showed that by 3 dpi, the negative strand RNLV1 RNA1 and RNA2 did accumulate in the Sf9 cells transfected with the in vitro transcribed viral RNAs or inoculated with the P1-supernatant from the transfected Sf9 cells (Figure 6b). The inoculated rice plants were harvested at two weeks post inoculation, and analyzed for virus infection through RT-PCR. The result showed that RNLV1 did not infect these rice plants (data not shown).

### 3.8. RNLV1 CP Is Self-Cleavable

CPs of alaphnodavirus are endopeptidases that can catalyze the hydrolysis of an asparaginyl bond at its C-terminus during virion maturation [1]. Self-cleavage of CP in provirions has been reported to be important for FHV infectivity [56]. The CP of RNLV1 was predicted to contain two peptidase family A21 domains other than the conserved peptidase family A6 domain. However, RNLV1 CP was also found to lack a conserved cleavage site N/A (alphanodavirus) or N/F or N/G (tetravirus) at its C terminus [1,46,47,57]. We suspected that the cleavage site might be N_480_/E at the C-terminus of RNLV1 CP through protein sequence alignments (Figure 7a). To determine whether RNLV1 CP can also function as an endopeptidase, we expressed His-tagged RNLV1 CP in *E. coli* BL21 cells followed by protein purification and polyclonal antibody production. We then expressed RNLV1 CP in *N. benthamiana* leaves through agro-infiltration or in Sf9 cells through transfection using a baculovirus-based vector. It was reported that the self-cleavage of Nudaurelia capensis omega virus (NωV) CP occurs in acidic environments [58,59]. Given that both NωV and RNLV1 CPs contain a peptidase A21 domain, we adjusted the pH value of the protein extract harvested from RNLV1 CP-expressed in *N. benthamiana* leaves or Sf9 cells (at 3 dpi.) to 5.0, and then incubated the extract for 1 d at room temperature. The resulting extract was analyzed through Western blot assay using the prepared RNLV1 CP-specific polyclonal antibody. Western blot assay found that the intact 58.5 kDa RNLV1 CP was degraded to produce a 54 kDa protein in acid environments (Figure 7b,c), which confirmed that RNLV1 CP^WT^ can self-cleave. We then expressed a mutant RNLV1 CP with a point mutation at asparagine 480 to alanine (referred to as CP^N480A^) in *N. benthamiana* leaves, after the same treatment in acidic environments, the resulting extract was also analyzed through Western blot assay using the RNLV1 CP-specific polyclonal antibody. The results showed no cleavage protein band in this mutant (Figure 7c), indicating N_480_/E at the C-terminus of RNLV1 CP is the cleavage site.

## 4. Discussion

In this study, we discovered and characterized two novel noda-like viruses in rice plants showing strong dwarf phenotypes, and named them as RNLV1 and RNLV2. Both viruses shared some characteristics with other nodaviruses, including their genome size, genome organization, and functions of viral proteins. According to the current classification criteria, these two new viruses should be placed in the family *Nodaviridae* based on the features of their RdRps. Viruses in the family *Nodaviridae* are currently divided into two genera according to their host range and CP characters [1,60]. However, many reported noda-like viruses in the family *Nodaviridae* cannot be placed in these two genera, due to their low sequence homologies with the known nodaviruses as well as their host ranges. 

Although genome organizations of tetraviruses are quite different, they encode closely related CPs (Appendix A) [47]. Tetraviruses are currently divided into three different families according to the latest ICTV Taxonomy [47]. In this study, we have found that RNLV1 CP contains two peptidase family A21 domains, similar to that of tetraviruses and two unclassified noda-like viruses (MoNV and LNV) (Appendix A), and can self-cleave at the N/E cleavage site at its C-terminus in acidic environments. We speculate that RNLV1 is likely originated from recombination between two viruses belonging to different families. A new genus in the family *Nodaviridae* may be required for the noda-like viruses encoding self-cleavage CPs with peptidase A21 domain(s).

RNLV2 encodes a CP without an endopeptidase domain, the same as some unclassified nodaviruses that have different hosts (Appendix A). Although RNLV2 is more closely related to betanodaviruses compared to alphanodaviruses according to their phylogenetic relationships, we cannot place it in the genus *Betanodavirus*, due mainly to its low CP sequence homology with that of the known betanodaviruses. The members in the genus *Betanodavirus* share over 80% CP aa sequence homology [1,60], while RNLV2 CP only shared ~20% aa sequence with that of betanodaviruses. Therefore, we propose to place RNLV2 in a new genus characterized by the CPs that have no self-cleavage activities. 

Movement protein (MP) is a viral protein that can assist virus intercellular and systemic movement in plants. It is known that FHV can replicate in plant cells but cannot move systematically in plants [61]. After FHV was inoculated to transgenic *N. benthamiana* plants expressing the MP of tobacco mosaic virus (TMV) or red clover necrotic mosaic virus (RCNMV), FHV moved systemically in the plants and accumulated in young non-inoculated leaves [62]. To investigate whether the proteins encoded by RNLV1 or RNLV2 can function as MP, we utilized potato virus X (PVX)-GFPΔp25, a movement-defective plant virus, that can replicate in plant cells but is defective in cell-to-cell movement [63]. It is reported that PVX-GFPΔp25 can recover cell-to-cell movement in *N. benthamiana* leaves expressed an MP from a different plant virus, such as sc4 of sonchus yellow net virus [64]. To test this possibility, we cloned RNLV1 and RNLV2 ORFs, individually, into the pGD expression vector and co-expressed them with the PVX-GFPΔp25 mutant in *N. benthamiana* leaves through agro-infiltration. By 6 dpi, green fluorescence was found again in single cells, the same as that found in the leaves expressing PVX-GFPΔp25 alone (data not shown), indicating that PVX-GFPΔp25 could replicate in a single plant cell but could not move between plant cells. These results showed that the proteins encoded by RNLV1 or RNLV2 could not complement the cell-to-cell movement of PVX-GFPΔp25 mutant.

It was reported that expression of the B2 protein of red-spotted grouper nervous necrosis virus (a betanodavirus) in fish cells induced hydrogen peroxide production and cell death [65,66]. Because RNLV1 and RNLV2 were found in rice plants showing strong dwarf phenotypes, we speculate that these two noda-like viruses can inhibit rice growth. Further investigations are needed to determine the potential damages of RNLV1 and RNLV2 to rice production. 

FHV has been shown to replicate in yeast, insect, plant, and mammalian cells [1,30,61,67]. In this study, we used different methods (i.e., mechanical inoculation and infectious clone inoculation) to test Koch’s Postulate and to determine if these two viruses can infect rice and *N. benthamiana* plants. Unfortunately, all the inoculated plants remained asymptomatic. Only RNLV1 RNA1 was found to replicate in *N. benthamiana* cells. We cloned the full-length RNLV1 RNA2 cDNA into the pCB301 vector and co-inoculated it with RNLV1 RNA1opt vector to *N. benthamiana*. Again, none of the inoculated plants showed virus symptoms. Western blot analysis showed that RNLV1 CP did accumulate in the inoculated *N. benthamiana* leaves, but not in their systematic leaves (data not shown). Therefore, whether RNLV1 and RNLV2 can infect whole plants remain unknown. It is possible that these two novel noda-like viruses are transmitted to rice plants in fields by unidentified insect(s). This insect vector(s) may also serve as the host(s) for virus replication. More field surveys and sample analyses are needed to address these unanswered questions.

Due to their relatively small genomes, alphanodaviruses, especially FHV and NoV, have been modified and used as models in many studies [28,29,30,31,32]. They have also been used as viral vectors in basic and applied research, including heterologous protein expression and gene silencing [1,68,69,70,71]. Because RNLV1 RNA1 can also replicate in plant cells and RNLV1 can infect insect Sf9 cells, we consider this virus as a potential vector for gene function studies in plants and insects.

In summary, we have identified two novel noda-like viruses (i.e., RNLV1 and RNLV2) in rice plants showing dwarf symptoms. Both viruses are clearly different from the reported nodaviruses, especially the CPs. These findings enrich our knowledge of nodaviruses and provide us with new virus-based vectors for future studies on the interaction between insect and plant. These findings can also help to develop more effective methods for rice virus disease management.

## Figures and Tables

**Figure 1 viruses-14-01159-f001:**
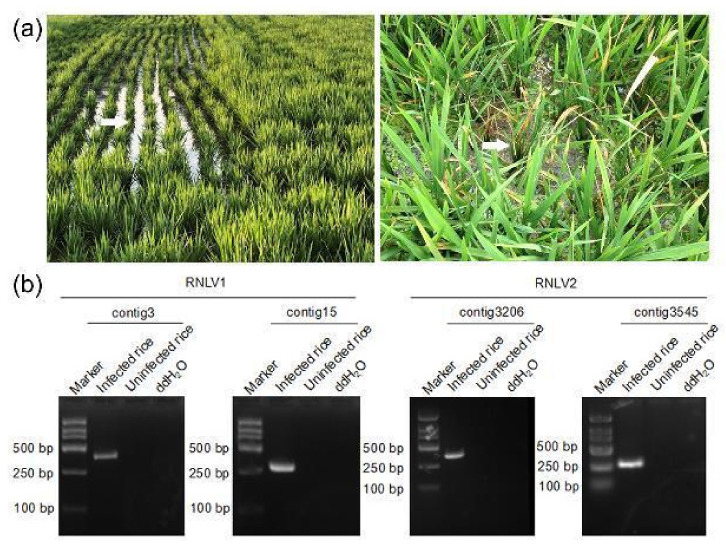
Images of dwarfed rice plants in fields and RT-PCR detection of virus infections. (**a**) Images of rice plants showing dwarf symptoms in a rice field in Shanghai City (left image) and in the Zhejiang Province (right image). White arrows indicate the plants showing dwarf symptoms. (**b**) RT-PCR detection of virus infection in the rice plants shown in (**a**) using primers designed according to the contig3 sequence (384 bp), the contig15 sequence (303 bp), the contig3206 sequence (366 bp), and the contig3545 sequence (279 bp), respectively.

**Figure 2 viruses-14-01159-f002:**
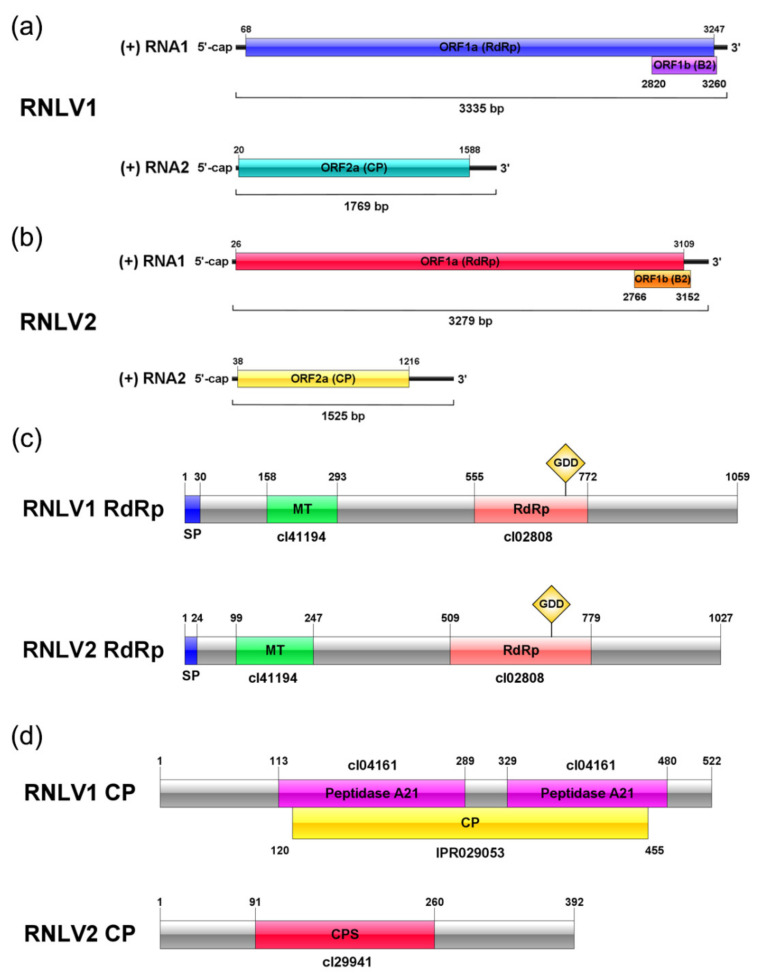
Genome organizations and protein domains of RNLV1 and RNLV2. (**a**) Genome organization of RNLV1. The predicted ORFs are shown in rectangles with different colors. (**b**) Genome organization of RNLV2. The predicted ORFs are shown in rectangles with different colors. (**c**) Two predicted domains in the RdRps of RNLV1 and RNLV2. The predicted domains are presented in rectangles with different colors. The domain accession numbers in CD database of the predicted domains are shown below the corresponding rectangles. SP, Signal peptide; MT, nodavirus vmethyltransferase domain; RdRp, RNA-dependent RNA polymerase domain; GDD, the conserved GDD box in the RdRp domain. (**d**) Predicted domains in the RNLV1 and RNLV2 CPs. The predicted CPs and domains are shown in rectangles with different colors. The domain accession numbers in CD and InterPro database are indicated above or below the corresponding rectangles. Peptidase A21, Peptidase family A21 domain; CP, coat protein domain; and CPS, coat protein s domain.

**Figure 3 viruses-14-01159-f003:**
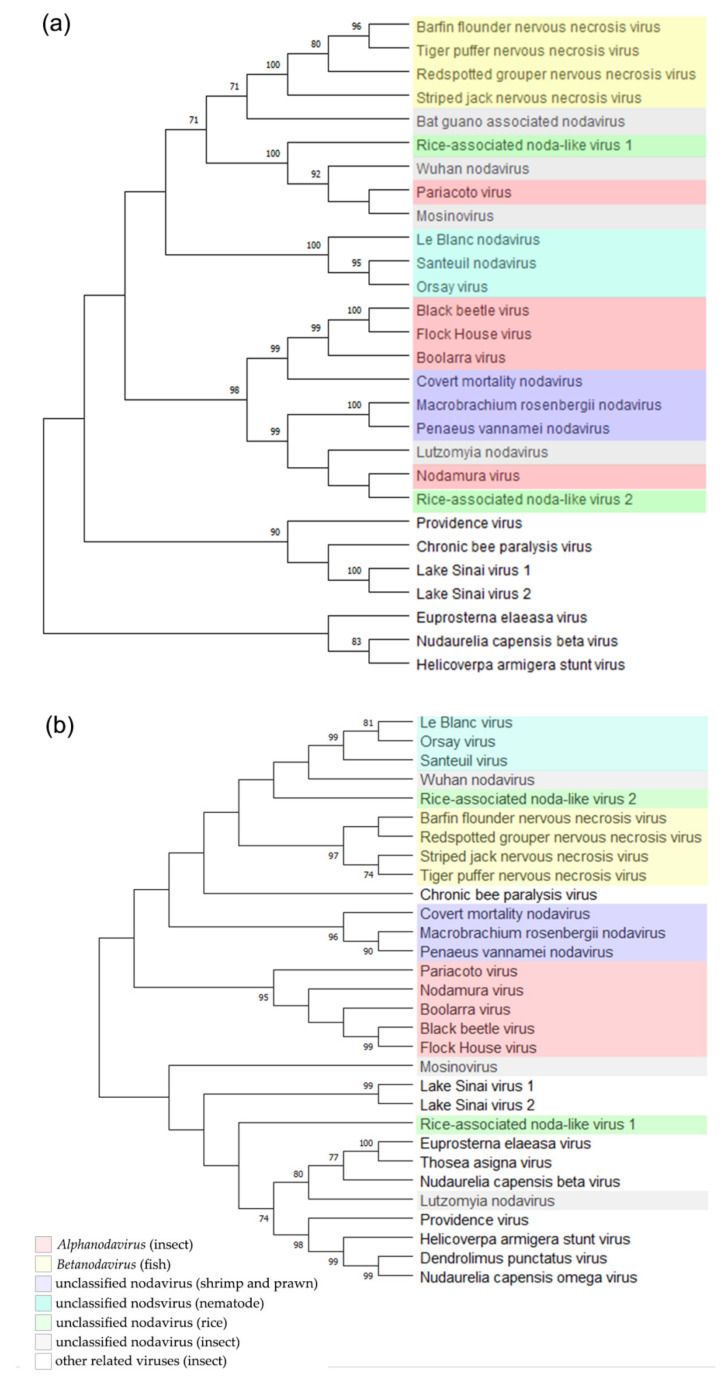
Phylogenetic relationships between RNLV1, RNLV2, and other related viruses. The phylogenetic trees were constructed using the RdRp sequences (**a**) or the CP sequences (**b**) and the maximum likelihood method with 1000 bootstraps. Accession numbers of these sequences are listed in Appendix A. The bootstrap values are indicated adjacent to the nodes. Taxonomy and hosts of these analyzed viruses are indicated.

**Figure 4 viruses-14-01159-f004:**
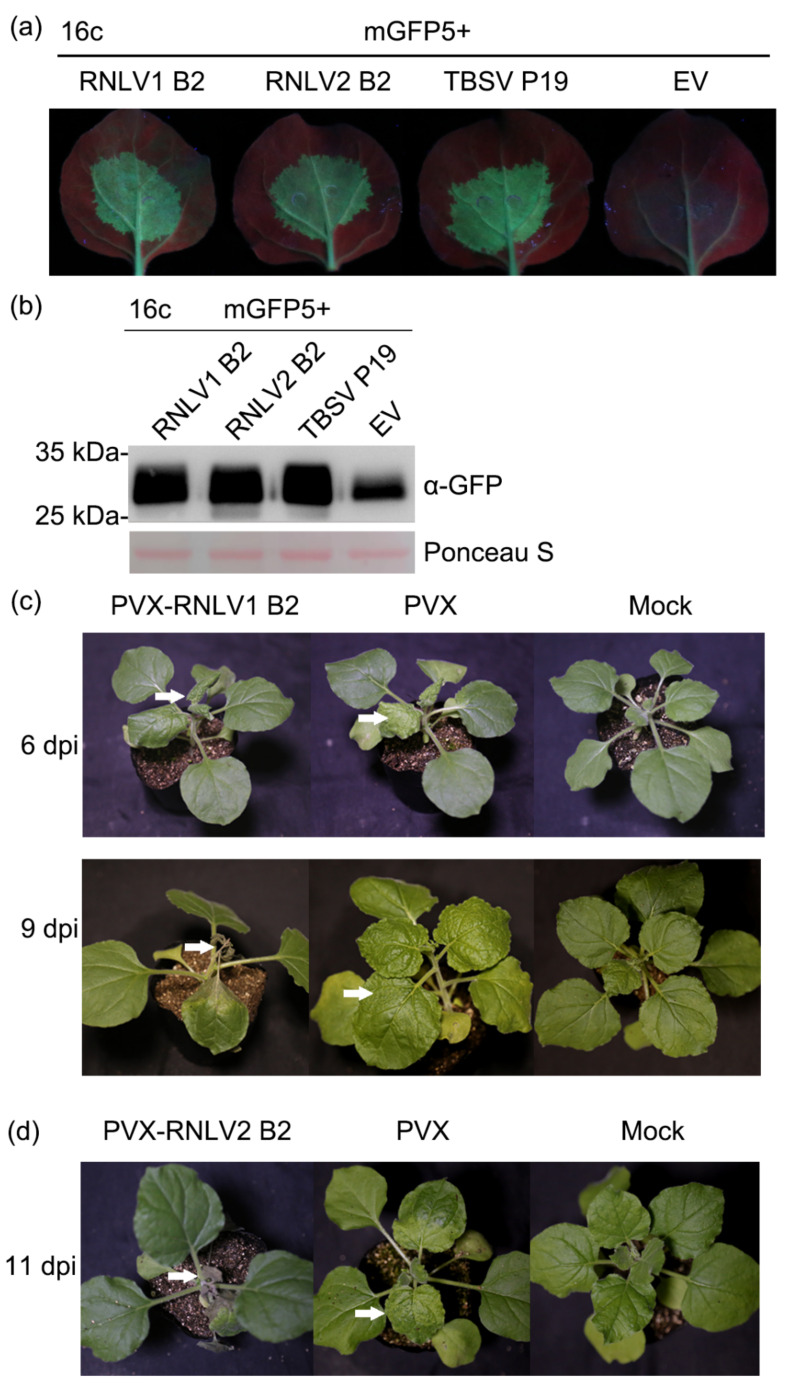
Both RNLV1 and RNLV2 B2 proteins can suppress RNA silencing in plants. (**a**) Images of transgenic 16c *N. benthamiana* leaves co-expressing mGFP5 and RNLV1 B2, mGFP5, and RNLV2 B2, mGFP5 and TBSV P19 (the positive control), or mGFP5 and the empty vector (EV, the negative control). These leaves were examined and photographed under a UV light at 5 dpi. Weak green fluorescence in the leaf tissues co-expressing mGFP5 and EV indicated the silencing of mGFP5 expression. (**b**) Western blot analysis of mGFP5 accumulation in the infiltrated leaf tissues at 5 dpi. The Ponceau S-stained large RuBisCO subunit gel is used to show sample loadings. (**c**) Images of PVX-RNLV1 B2-, PVX-, and mock-inoculated *N. benthamiana* plants at 6 and 9 dpi. White arrows indicate the leaves showing disease symptoms. (**d**) By 11 dpi, the PVX-RNLV2 B2-inoculated plants also developed necrosis in their systemic leaves, but not the PVX-inoculated or the mock-inoculated *N. benthamiana* plants. White arrows indicate the leaves showing disease symptoms.

**Figure 5 viruses-14-01159-f005:**
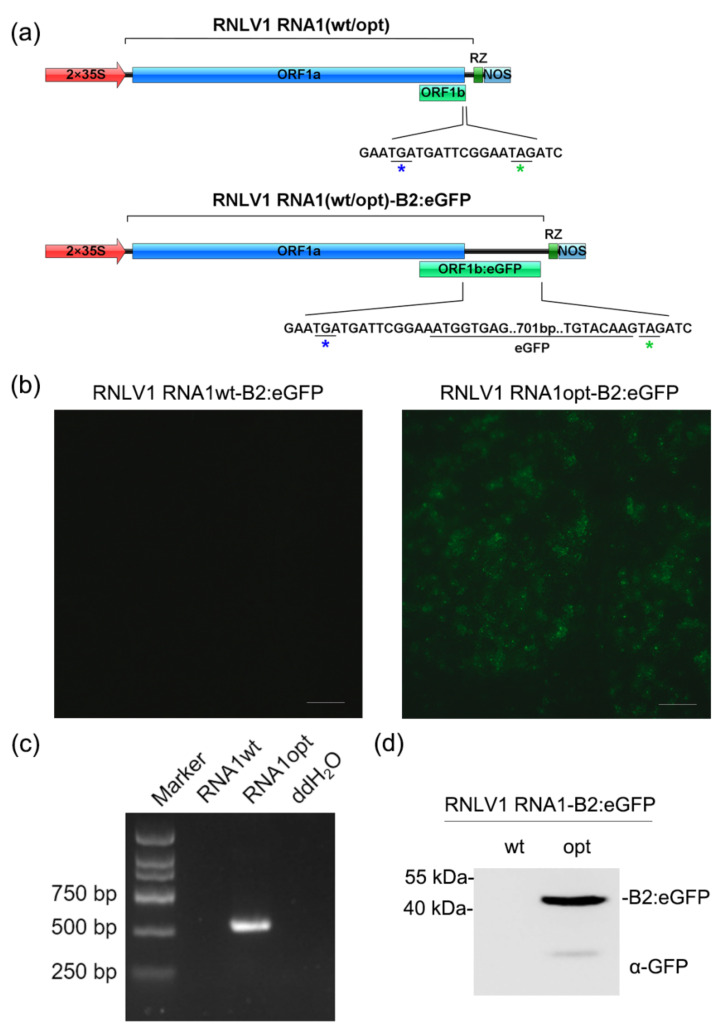
RNLV1 RNA1 can replicate in *N. benthamiana* leaf cells. (**a**) Insertion site of an *eGFP* gene in RNLV1 RNA1 in the pCB301-RNLV1 RNA1(wt/opt)-B2:eGFP vector. 2 × 35S, doubled CaMV 35S promoter; RZ, HDV ribozyme; NOS, NOS terminator. (**b**) The infiltrated *N. benthamiana* leaves were photographed at 3 dpi under a fluorescence microscope. (**c**) RT-PCR analysis of RNLV1 RNA1 replication in the infiltrated leaf tissues shown in (**b**) using primers specific for RNLV1 RNA1wt or RNA1opt (560 bp). (**d**) Analysis of fusion protein accumulation in the infiltrated leaf tissues shown in (**b**) through Western blot assay using a GFP specific antibody at 3 dpi.

**Figure 6 viruses-14-01159-f006:**
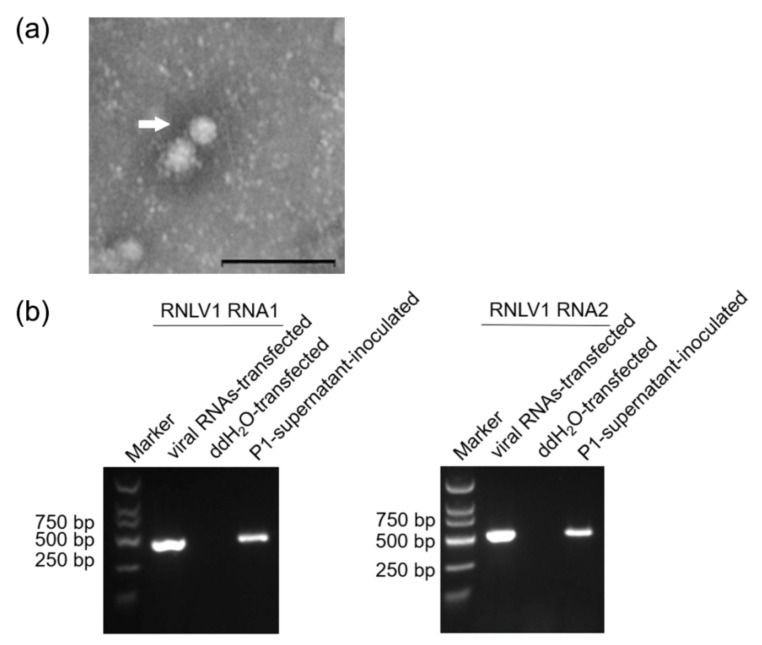
RNLV1 can replicate in insect Sf9 cells. (**a**) An electron micrograph showing negatively stained RNLV1 virions from the transfected Sf9 cells. Bars = 200 nm. (**b**) RT-PCR analysis of RNLV1 replication in Sf9 cells. Sf9 cells were transfected with in vitro transcribed RNLV1 RNA transcripts, double distilled water (ddH_2_O, the negative control), or inoculated with the P1-supernatant from the transfected Sf9 cells. After three days, these cells were analyzed for RNLV1 infection through RT-PCR using primers specific for the negative strand RNA1 (422 bp, left panel) or RNA2 (509 bp, right panel).

**Figure 7 viruses-14-01159-f007:**
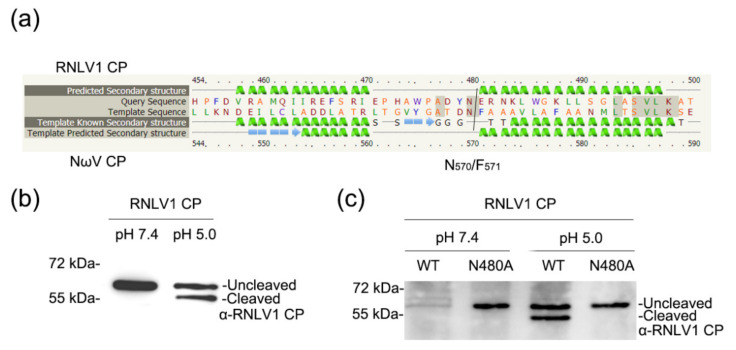
Analysis of RNLV1 CP self-cleavage. (**a**) Sequence alignments of RNLV1 CP and NωV CP using Phyre2 service. (**b**) Western blot assay analyzing the self-cleavage of RNLV1 CP from the transfected Sf9 cells. Extracts were incubated in a weakly alkaline (pH 7.4) or acidic environment (pH 5.0) at room temperature for 1 d and then analyzed through Western blot assay using a polyclonal antibody against RNLV1 CP. (**c**) Western blot assay analyzing the self-cleavage of RNLV1 CP^WT^ and the mutant RNLV1 CP^N480A^ from *N. benthamiana* leaves agro-infiltrated with pGD-RNLV1 CP^WT^ and pGD-RNLV1 CP^N480A^. At 3 dpi, extracts were incubated in a weakly alkaline (pH 7.4) or acidic environment (pH 5.0) at room temperature for 1 d and then analyzed through Western blot assay using a polyclonal antibody against RNLV1 CP.

**Table 1 viruses-14-01159-t001:** Terminal sequences of RNLV1 RNA1 and RNA2.

RNLV1	Methods	5′ and 3′-Terminal Sequence		Clones
RNA1	5′-RACE	GAUAUAUUAUC...		6/6 ^1^
	3′-RACE		...UAACACCGGCG	4/6
			...UAACACCGGGG	2/6
	junction RT-PCR	AUAUAUUAUC...	...UAACACCGGC ^2^	6/6
RNA2	5′-RACE	GAUAUAUUAUC...		6/6
	3′-RACE		...AUGUACUGGCG	5/5
	junction RT-PCR	AUAUAUUAUC...	...AUGUACUGGC	6/6

^1^ The number of clones had the sequence shown versus the total number of clones analyzed. ^2^ Sequences obtained by junction RT-PCR are separated into the corresponding 5′- and 3′-termini for clarity.

**Table 2 viruses-14-01159-t002:** Terminal sequences of RNLV2 RNA1 and RNA2.

RNLV2	Methods	5′ and 3′-Terminal Sequence		Clones
RNA1	5′-RACE	GGUUUUGUUAC...		6/6 ^1^
	3′-RACE		...UUUAGAACGGU(A)	5/9
			...UUUAGAACGU(A)	4/9
RNA2	5′-RACE	GGUAUUGUUUU...		6/6
	3′-RACE		...AGAAAUUUCU(A)	5/5

^1^ The number of clones had the sequence shown versus the total number of clones analyzed.

**Table 3 viruses-14-01159-t003:** Sequence homologies between RNLV1 genomic RNAs, proteins and that of other nodaviruses.

Genus	Virus Species	RNA1	RNA2	RdRp	B2	CP
*Alphanodavirus*	Black beetle virus	39.58% ^1^	32.02% ^1^	19.85% ^2^	9.59% ^2^	14.83% ^2^
*Alphanodavirus*	Boolarra virus	37.37%	30.19%	19.89%	15.07%	10.15%
*Alphanodavirus*	Flock House virus	39.87%	30.58%	19.95%	9.59%	14.45%
*Alphanodavirus*	Nodamura virus	38.49%	31.80%	19.28%	14.19%	12.57%
*Alphanodavirus*	Pariacoto virus	42.33%	29.04%	33.05%	13.01%	12.95%
*Betanodavirus*	Barfin flounder nervous necrosis virus	39.35%	34.21%	20.94%	8.90%	11.11%
*Betanodavirus*	Redspotted grouper nervous necrosis virus	39.36%	33.98%	21.59%	10.96%	11.11%
*Betanodavirus*	Striped jack nervous necrosis virus	40.06%	32.64%	21.26%	10.27%	10.15%
*Betanodavirus*	Tiger puffer nervous necrosis virus	39.82%	30.99%	21.11%	9.59%	10.54%
Unclassified	Macrobrachium rosenbergii nodavirus	40.02%	29.07%	19.20%	8.72%	12.81%
Unclassified	Penaeus vannamei nodavirus	40.03%	27.62%	19.53%	12.84%	10.71%
Unclassified	Covert mortality nodavirus	39.53%	31.77%	18.67%	17.01%	12.81%
Unclassified	Wuhan nodavirus (Pieris rapae virus)	42.89%	39.04%	30.65%	13.84%	10.54%
Unclassified	Le Blanc virus	39.92%	21.06%	15.53%	- ^3^	9.90%
Unclassified	Santeuil virus	37.64%	28.91%	15.83%	-	8.99%
Unclassified	Orsay virus	40.20%	28.26%	16.47%	-	8.43%
Unclassified	Mosinovirus	43.57%	39.62%	36.27%	10.27%	16.42%
Unclassified	Lutzomyia nodavirus	38.74%	36.66%	17.31%	15.20%	13.99%
Unclassified	Bat guano associated nodavirus	38.69%	-	25.54%	-	-

^1^ nucleotide sequence similarities between RNLV1 RNA1, RNA2, and that of other nodaviruses, determined using the DNAMAN software. ^2^ amino acid sequence identities between the predicted RNLV1 proteins and that of other nodaviruses, determined using the DNAMAN software. ^3^ sequences of viruses were not available.

**Table 4 viruses-14-01159-t004:** Sequence homologies between RNLV2 genomic RNAs, proteins and that of other nodaviruses.

Genus	Virus Species	RNA1	RNA2	RdRp	B2	CP
*Alphanodavirus*	Black beetle virus	47.12% ^1^	38.40% ^1^	42.56% ^2^	21.88% ^2^	12.80% ^2^
*Alphanodavirus*	Boolarra virus	48.26%	33.96%	39.34%	20.31%	12.71%
*Alphanodavirus*	Flock House virus	47.30%	37.82%	40.43%	19.53%	12.20%
*Alphanodavirus*	Nodamura virus	52.33%	34.06%	47.76%	26.81%	13.45%
*Alphanodavirus*	Pariacoto virus	40.08%	35.98%	19.55%	12.50%	12.96%
*Betanodavirus*	Barfin flounder nervous necrosis virus	40.31%	36.64%	22.38%	10.16%	21.68%
*Betanodavirus*	Redspotted grouper nervous necrosis virus	41.62%	36.71%	22.26%	10.94%	21.43%
*Betanodavirus*	Striped jack nervous necrosis virus	40.30%	39.65%	22.87%	11.72%	19.90%
*Betanodavirus*	Tiger puffer nervous necrosis virus	40.59%	37.45%	22.40%	13.28%	20.92%
Unclassified	Macrobrachium rosenbergii nodavirus	49.73%	32.08%	43.67%	22.86%	13.42%
Unclassified	Penaeus vannamei nodavirus	49.97%	31.90%	43.12%	21.43%	13.35%
Unclassified	Covert mortality nodavirus	46.49%	39.58%	36.65%	22.96%	11.36%
Unclassified	Wuhan nodavirus (Pieris rapae virus)	40.04%	36.14%	18.85%	12.50%	15.75%
Unclassified	Le Blanc virus	38.71%	18.13%	15.58%	- ^3^	15.31%
Unclassified	Santeuil virus	36.72%	24.66%	14.94%	-	15.46%
Unclassified	Orsay virus	38.79%	24.66%	14.91%	-	15.75%
Unclassified	Mosinovirus	37.78%	34.21%	21.16%	10.94%	11.69%
Unclassified	Lutzomyia nodavirus	47.90%	28.86%	38.33%	13.10%	8.16%
Unclassified	Bat guano associated nodavirus	38.86%	-	21.42%	-	-
Unclassified	Rice-associated noda-like virus 1	39.89%	35.24%	17.92%	17.12%	10.92%

^1^ nucleotide sequence similarities between RNLV2 RNA1, RNA2, and that of other nodaviruses, determined using the DNAMAN software. ^2^ amino acid sequence identities between the predicted RNLV2 proteins and that of other nodaviruses, determined using the DNAMAN software. ^3^ sequences of viruses were not available.

## Data Availability

Not applicable.

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
