# Peer review of "Identification and Characterization of Two Novel Noda-like Viruses from Rice Plants Showing the Dwarfing Symptom"

_viruses, 2022, doi:10.3390/v14061159_

Round 1

Reviewer 1 Report

The manuscript by Xie et al. describes the identification of two new viruses, RNLV1 and RNLV2, from rice exhibiting symptoms of dwarfing through the use of high-throughput sequencing (HTS), their full molecular characterization, and establishment of a manipulatable genetic system for each of the viruses by creating two infectious clones. This is a nice project that presents novel and significant findings about two new viruses infecting rice. The manuscript is well-written and does not need any editing. There are, however, some relatively minor issues that need to be addressed, related to designations of the conserved domains in the virus polyproteins, and some typing errors.

This reviewer understands that the authors used the domain designations from a specific program available at the NCBI web-site, but these were not intended for any specific use for a particular virus – they represent generic definitions across the multitude of proteins from different systems, and need to be adapted for this specific situation. Obviously, nodaviruses are not retroviruses and cannot have a reverse transcriptase domain in their polyproteins. It is the RdRp which has evolutionary relationship to the RT. The same can be said about the methyltransferase domain that happens to be designated in the CD Search program as ‘vmethyltransferase’, to separate it from other, cellular methyltransferases. Hence, there is no need to use exactly the same designation ‘vmethyltransferase’ here, the reader knows already that we are talking about a virus. The recommendation is to use RdRp throughout the manuscript, including the text and the figures, replacing the confusing and incorrect ‘RT’, and also to use ‘methyltransferase or MT’ designation as well.   

Specific points to address:

  1. l. 17 – this sentence needs re-phrasing, since ‘zoonotic’ has a narrow meaning describing animal infections transmissible to humans; better state that nodaviruses have not been found in plant hosts up to now
  2. l. 26 – here and throughout the manuscript, B1 and B2 genes/proteins are mentioned but these two designations could not be found on any genome diagram, making unclear what open reading frames encode these two proteins
  3. l. 317 – correct the title of the table, this is RNLV2 presented in the table
  4. l. 326 – correct ‘methyltransferase domain’
  5. l. 326 – replace with ‘RdRp’ domain, since this is not a retrovirus

Fig. 2 – note that nowhere on this figure you can find the B2 label, yet in the text it is discussed quite extensively

Fig. 3 – in the phylogenies, all nodes with bootstrap values under 70% should be collapsed

  1. l. 419 – correct ‘species’
  2. l. 467 – note that this is the first instance where B2 is disclosed as encoded by ORF1b, on page 16 of the manuscript!

Author Response

Point-by-point responses to the reviewers’ comments, and a list of changes is as follows:

Reviewer 1#

The manuscript by Xie et al. describes the identification of two new viruses, RNLV1 and RNLV2, from rice exhibiting symptoms of dwarfing through the use of high-throughput sequencing (HTS), their full molecular characterization, and establishment of a manipulatable genetic system for each of the viruses by creating two infectious clones. This is a nice project that presents novel and significant findings about two new viruses infecting rice. The manuscript is well-written and does not need any editing. There are, however, some relatively minor issues that need to be addressed, related to designations of the conserved domains in the virus polyproteins, and some typing errors.

Our response: We would like to thank the respected reviewer 1 for your positive comments and valuable suggestions.

This reviewer understands that the authors used the domain designations from a specific program available at the NCBI web-site, but these were not intended for any specific use for a particular virus – they represent generic definitions across the multitude of proteins from different systems, and need to be adapted for this specific situation. Obviously, nodaviruses are not retroviruses and cannot have a reverse transcriptase domain in their polyproteins. It is the RdRp which has evolutionary relationship to the RT. The same can be said about the methyltransferase domain that happens to be designated in the CD Search program as ‘vmethyltransferase’, to separate it from other, cellular methyltransferases. Hence, there is no need to use exactly the same designation ‘vmethyltransferase’ here, the reader knows already that we are talking about a virus. The recommendation is to use RdRp throughout the manuscript, including the text and the figures, replacing the confusing and incorrect ‘RT’, and also to use ‘methyltransferase or MT’ designation as well.

Our response: We would like to thank the respected reviewer 2 for your valuable suggestions. In the revised manuscript, we have used RdRp throughout the manuscript including the text and the figures, instead of ‘RT’, and also to use ‘methyltransferase or MT’ designation as well.

Specific points to address:

  1. 17 – this sentence needs re-phrasing, since ‘zoonotic’ has a narrow meaning describing animal infections transmissible to humans; better state that nodaviruses have not been found in plant hosts up to now

Our response: Revised accordingly.

  1. 26 – here and throughout the manuscript, B1 and B2 genes/proteins are mentioned but these two designations could not be found on any genome diagram, making unclear what open reading frames encode these two proteins

Our response: We have improved Figure 2 and FigureS 3 via matching the protein names with the encoding ORFs.

  1. 317 – correct the title of the table, this is RNLV2 presented in the table.

Our response: Revised accordingly.

  1. 326 – correct ‘methyltransferase domain’

Our response: Revised accordingly.

  1. 326 – replace with ‘RdRp’ domain, since this is not a retrovirus

Our response: Revised accordingly.

  1. 2 – note that nowhere on this figure you can find the B2 label, yet in the text it is discussed quite extensively

Our response: Revised accordingly.

  1. 3 – in the phylogenies, all nodes with bootstrap values under 70% should be collapsed

Our response: Revised accordingly.

  1. 419 – correct ‘species’

Our response: Revised accordingly.

  1. 467 – note that this is the first instance where B2 is disclosed as encoded by ORF1b, on page 16 of the manuscript!

Our response: In fact, we mentioned B2 when we describe the viral genome on page 8. In this revised manuscript, we have made some changes in both text and figures for better understanding the correspondence between the ORFs and the encoding proteins.

Reviewer 2 Report

In this manuscript Xie and co-authors identified and characterized two novel noda-like viruses from rice. The authors performed robust experiments and well presented to complete the characterization of these new RNLV1 and RNLV2 viruses.

I think that this manuscript is suitable for the journal viruses.

Author Response

Reviewer 2#

This manuscript reported a rice infecting nodavirus, which is not reported in rice yet. Its close relative FHV was reported to infect plants systemically in presence of heterologous virus movement protein expressed as transgene. But no natural nodavirus infecting plants was confirmed. Since Nodavirus is a well established model insect virus. The reported virus would be possible to help establish a good virus-plant-insect trilateral interaction system. This study is of strong interest in plant-virus interaction and in crop protection.

Our response: We would like to thank the respected reviewer 2 for your positive comments.

Reviewer 3 Report

Please check the following suggestions for the improvement.

Line 102-103, Fig.1a

Distribution pattern of the dwarfing rice plants in Shanghai is very uniform, and they seem unlikely to be transmitted by insect vectors. Anyway, I think readers are interested in the scale of occurrence of these symptoms, although these viruses did not fulfil Koch's postulate. Please describe how much area of paddy fields suffered the damages of these symptoms. Did they occur in only one paddy field respectively, multiple paddy fields, or in a large area?

Fig.2

The indication letters are wrong. Please add signs of (a), (b), (c), (d) from the top of the figure correctly.

Line 450-451

I think that it seems like a leap in logic to conclude that B2 is definitely a symptom determinant, based on the fact that the B2 protein suppressed silencing of PVX symptom expression. Isn't it better to describe that it is involved in or related to symptom determinant?  Please consider it.

Author Response

Reviewer 3#

In this manuscript Xie and co-authors identified and characterized two novel noda-like viruses from rice. The authors performed robust experiments and well presented to complete the characterization of these new RNLV1 and RNLV2 viruses.I think that this manuscript is suitable for the journal viruses.

Our response: We would like to thank the respected reviewer 3 for your positive comments.

Please check the following suggestions for the improvement.

  1. Line 102-103, Fig.1a – Distribution pattern of the dwarfing rice plants in Shanghai is very uniform, and they seem unlikely to be transmitted by insect vectors. Anyway, I think readers are interested in the scale of occurrence of these symptoms, although these viruses did not fulfil Koch's postulate. Please describe how much area of paddy fields suffered the damages of these symptoms. Did they occur in only one paddy field respectively, multiple paddy fields, or in a large area?

Our response: We would like to thank the respected reviewer 3for your valuable suggestions. According your suggestion, this revised manuscript is added the content on page 5: In 2019, during a survey of rice viral diseases in Shanghai City, China, we found rice plants showing similar dwarfing symptoms in a paddy field of about one hectare (Figure 1A, left). In 2016, during rice field survey in Zhejiang Province, China, we found some dwarf rice plants in multiple paddy fields of approximately twenty hectares (Figure 1A, right).

  1. 2 – The indication letters are wrong. Please add signs of (a), (b), (c), (d) from the top of the figure correctly.

Our response: Revised accordingly.

  1. Line 450-451 – I think that it seems like a leap in logic to conclude that B2 is definitely a symptom determinant, based on the fact that the B2 protein suppressed silencing of PVX symptom expression. Isn't it better to describe that it is involved in or related to symptom determinant?  Please consider it.

Our response: Plants inoculated with PVX-based vector expressing strong suppressors often show more severe symptoms, we agree that it’s not appropriate to suggest that B2 is a symptom determinant based on the results in the article, so we rephrased it as B2 is related to symptom determinant.